# Tspan8 Drives Melanoma Dermal Invasion by Promoting ProMMP-9 Activation and Basement Membrane Proteolysis in a Keratinocyte-Dependent Manner

**DOI:** 10.3390/cancers12051297

**Published:** 2020-05-21

**Authors:** Manale El Kharbili, Muriel Cario, Nicolas Béchetoille, Catherine Pain, Claude Boucheix, Françoise Degoul, Ingrid Masse, Odile Berthier-Vergnes

**Affiliations:** 1Centre de Génétique et de Physiologie Moléculaires et Cellulaires, CNRS UMR5534, Université de Lyon, F-69003 Lyon, France; manale.elkharbili@cuanschutz.edu (M.E.K.); o.vergnes@icloud.com (O.B.-V.); 2Department of Dermatology, University of Colorado Anschutz Medical Campus, Aurora, CO 80045, USA; 3National Reference Center for Rare Skin Disease, Department of Dermatology, University Hospital, INSERM 1035, F-33000 Bordeaux, France; muriel.cario-andre@u-bordeaux.fr (M.C.); paincatherine4@gmail.com (C.P.); 4AquiDerm, University Bordeaux, F-33076 Bordeaux, France; 5R&D Department, Gattefossé, F-69800 Saint-Priest, France; nbechetoille@gattefosse.com; 6INSERM U935, Université Paris-Sud, F-94800 Villejuif, France; claude.boucheix@inserm.fr; 7INSERM U1240, Université Clermont Auvergne, Imagerie Moléculaire et Stratégies Théranostiques, F-63000 Clermont Ferrand, France; francoise.degoul@inserm.fr; 8Centre de Recherche en Cancérologie de Lyon, CNRS-UMR5286, INSERM U1052, Université de Lyon, F-69008 Lyon, France; 9US7INSERM /UMS3453 UCBL SFR Santé Lyon-Est, F-69372 Lyon, France

**Keywords:** tumor microenvironment, melanoma invasion, melanoma-keratinocytes crosstalk, dermal-epidermal junction, dermis, Tetraspanin 8, MMP-9

## Abstract

Melanoma is the most aggressive skin cancer with an extremely challenging therapy. The dermal-epidermal junction (DEJ) degradation and subsequent dermal invasion are the earliest steps of melanoma dissemination, but the mechanisms remain elusive. We previously identified Tspan8 as a key actor in melanoma invasiveness. Here, we investigated Tspan8 mechanisms of action during dermal invasion, using a validated skin-reconstruct-model that recapitulates melanoma dermal penetration through an authentic DEJ. We demonstrate that Tspan8 is sufficient to induce melanoma cells’ translocation to the dermis. Mechanistically, Tspan8^+^ melanoma cells cooperate with surrounding keratinocytes within the epidermis to promote keratinocyte-originated proMMP-9 activation process, collagen IV degradation and dermal colonization. This concurs with elevated active MMP-3 and low TIMP-1 levels, known to promote MMP-9 activity. Finally, a specific Tspan8-antibody reduces proMMP-9 activation and dermal invasion. Overall, our results provide new insights into the role of keratinocytes in melanoma dermal colonization through a cooperative mechanism never reported before, and establish for the first time the pro-invasive role of a tetraspanin family member in a cell non-autonomous manner. This work also displays solid arguments for the use of Tspan8-blocking antibodies to impede early melanoma spreading and therefore metastasis.

## 1. Introduction

Cutaneous melanoma is the deadliest skin cancer due to its high metastatic propensity and resistance to most conventional and targeted therapies [1]. It usually progresses from an early radial growth phase (RGP) confined within the epidermis to a vertical growth phase (VGP) characterized by dermal invasion, where metastasis risk is high [2]. To date, Breslow thickness remains the most powerful prognostic factor, as long as metastases are not present at the time of diagnosis. The most recent AJCC 8th guidelines introduced mitotic rate as an additional criterion for thinner melanomas, the presence of >1 mitosis/mm² predicts poorer outcome [3]. Moreover, the ulceration status used for the sub-classification of thin melanomas [3] emerges as another important histological factor predicting survival [4]. However, such histological features define prognostic groups but not individual patient risk. Indeed, even though the survival rate for thin melanomas is high, some patients develop metastases [3]. Moreover, Werner-Klein et al. [5] recently showed that dissemination occurs shortly after dermal invasion at a median tumor thickness of ~0.5 mm. Therefore, understanding the mechanisms that convert RGP melanoma into VGP is crucial to identify reliable predictive biomarkers and novel therapeutic targets. 

Cutaneous melanomas are composed of genotypically and phenotypically distinct subpopulations, dynamically regulated by the selective pressure imposed from the host tumor microenvironment and host immune system [6]. This tumor heterogeneity contributes largely to their strong resistance to standard, targeted and immune therapies [7,8]. Indeed, it appears that cancer/immune cell interactions are informative of resistance to immunotherapy whereas cancer/stromal cell interactions are informative of MAPK inhibitors’ resistance [9]. Consistently with the high inter- and intra-tumoral heterogenity of cutaneous melanomas, we have previously defined a subset of melanoma cells expressing strong levels of peanut agglutinin-receptors that possesses a high metastatic frequency [10] and correlates with poor patient survival [11], which simultaneously express Tetraspanin 8 (Tspan8) [12]. Tspan8 belongs to a four-transmembrane-domain protein family called tetraspanins, that organize membrane microdomains via interactions with other tetraspanins and a variety of transmembrane/cytosolic proteins to regulate a wide range of cellular functions, including proliferation, motility, metastasis and angiogenesis [13,14]. Tspan8 is categorized as pro-metastatic in various carcinomas [14] and emerged as an attractive therapeutic target [15,16] and a blood biomarker [17].

We were the first to reveal that Tspan8 expression is sufficient to transform non-invasive melanoma cells into invasive cells [12]. Tspan8 is undetectable at both mRNA and protein in healthy skin, but its expression is acquired by aggressive primary melanomas and lymph node metastases. We also demonstrated that *TSPAN8* is under the transcriptional control of LCMR1 and p53 [18,19] and acts not only by reducing matrix adherence via the β1-integrin/ILK signaling pathway [20], but also by promoting invasion through β-catenin activation [21]. 

It is accepted that reciprocal stroma–tumor interactions contribute to metastatic progression, especially through the production of matrix degrading enzymes such as MMPs [22,23]. However, the exact mechanisms governing the interplay between melanoma cells and epidermal microenvironment in controlling MMP-dependent invasion have not been studied to date. Here, we address how Tspan8 participates in the dermal–epidermal junction (DEJ) proteolysis during melanoma invasion and whether it contributes to tumor–keratinocyte crosstalk. To this aim, we used 3D-skin reconstructs (SR) with an authentic DEJ, which recapitulate early melanoma stages [24,25]. We found that mere Tspan8 gain of expression is sufficient to promote melanoma invasive behavior and acts by driving proMMP-9 activation leading to DEJ proteolysis. More importantly, we showed that Tspan8 function hinges on the dialog between tumor cells and neighboring keratinocytes. Our work provides strong evidence of the primary involvement of Tspan8 in melanoma–keratinocyte crosstalk leading to efficient DEJ degradation. This is, to our knowledge, the first report demonstrating bidirectional interplay between melanoma cells and epidermal microenvironment to regulate MMP-dependent invasion. This is also the first study characterizing the role of a tetraspanin family member in a cell non-autonomous mechanism that controls basement membrane proteolysis and local invasion.

## 2. Results

### 2.1. Tspan8 is Exclusively Expressed in the In Vivo-Selected Highly Metastatic and Invasive Melanoma Subsets

We previously developed an orthotopic rat model for the spontaneous metastasis of human melanoma [10]. This model allowed the selection from a non-aggressive parental cell line of subpopulations with low (NM#1, NM#2, NM#3) or high (M#1, M#2, M#3) lung metastatic potential. Figure 1a depicts a schematic of the selection procedures. M#1, M#2 and M#3 subsets expressed Tspan8 at the mRNA (Figure 1b), protein (Figure 1c), cell-surface (Figure 1d) levels, and displayed a high ability to invade Matrigel (Figure 1e), unlike the parental line and the non-metastatic NM#1, NM#2, NM#3 subsets. These results showed that the parental line is populated by melanoma cells with heterogeneous metastatic phenotypes and that Tspan8 is strongly expressed in the invasive/metastatic subsets. 

### 2.2. Tspan8 Expression in Melanoma Cells Promotes ProMMP-9 Activation, Collagen IV Degradation and DEJ Crossing

We next determined how Tspan8 expression affects dermal invasion. We used the SR, previously described [24,25] to accurately recapitulate the early steps of melanoma invasion through a preserved 3D architecture of native DEJ. After 21 days of culture, Tspan8^+^ cells invaded the dermis and formed numerous compact nodules (Figure 2a). By contrast, large nests of Tspan8^−^ cells were located exclusively in the epidermis, along the DEJ. Evidence of collagen IV dissolution, the major DEJ component, was observed exclusively when Tspan8^+^ cells were used and integrated within the epidermis (Figure 2b). These data demonstrate that keratinocytes are required for the penetration of Tspan8^+^ cells across the DEJ, and local degradation of collagen IV, both occurring around day 21. 

Collagen IV is known to be primarily degraded by MMP-9 and MMP-2. Therefore, we examined whether Tspan8 regulates MMP-9 and MMP-2 expression and/or activity by using zymographs and ELISA assays on conditioned media harvested from SR. A time-course study revealed that proMMP-9 became active as early as day 10 of culture and drastically increased until reaching active MMP-9 highest amount at the time of collagen IV dissolution (day 21), exclusively in medium from SR containing Tspan8^+^ cells (Figure 2c,d). Indeed, SR integrating Tspan8^−^ cells were capable of producing proMMP-9, but unable to generate its active form (Figure 2c,d), in accordance with the intact collagen IV layer (Figure 2b). 

Strikingly, melanoma cells were unable to cross the DEJ when keratinocytes were not incorporated, regardless of their Tspan8 expression levels (Figure 2a), even after 5 weeks of culture (not shown). This was consistent with the absence of breaks in collagen IV staining (Figure 2b), low levels of proMMP-9 and absence of active MMP-9 (Figure 2c,d). The MMP-2 proform was detected at very low levels without noticeable active MMP-2 irrespective of Tspan8 expression, at all times and in all tested culture conditions (Figure 2c,e). These data demonstrate that Tspan8 is a key determinant in the activation process of MMP-9, but not of MMP-2, which depends heavily on surrounding keratinocytes.

To further confirm that Tspan8 confers MMP-9-dependent invasive activity, we have generated stable clones expressing ectopic Tspan8 or depleted of endogenous Tspan8. We confirmed the efficiency of Tspan8 expression/silencing at the mRNA (Figure 3a), protein (Figure 3b) and cell-surface (Figure 3c) levels. We observed that non-invasive melanoma cells gained strong invasive properties after the ectopic expression of Tspan8 in matrigel (Figure 3d) and SR (Figure 3e) concomitantly to the production of high levels of active MMP-9 (Figure 3f). Conversely, Tspan8 depletion in invasive cells strongly inhibited matrigel invasion (Figure 3d) and efficiently prevented melanoma cells from crossing the DEJ (Figure 3e), concurrently to a drastic decrease in proMMP-9 activation (Figure 3f). Overall, these data show that, by itself, Tspan8 expression was sufficient to trigger proMMP-9 activation process and collagen IV dissolution, allowing melanoma cells to cross DEJ and invade dermis.

### 2.3. Tspan8^+^ Melanoma Cells Require Neighboring Keratinocytes to Promote Dermal Invasion

Tspan8^+^ cells crossed DEJ exclusively when integrated into an epidermal microenvironment (Figure 2a). We thus investigated whether and how keratinocytes influence melanoma invasion. We developed four different models schematized in Figure 4a. Keratinocytes and Tspan8^+^ melanoma cells were either cultured alone on de-epidermized dermis (DED) (I and II cultures, respectively) or cocultured, without or with cell–cell contacts (III and IV cultures, respectively). As shown in Figure 4b, Tspan8^+^ cells were able to penetrate the DEJ only when surrounded by keratinocytes, which coincided with collagen IV breakdown (Figure 4c). Strikingly, when cultured alone or cocultured with keratinocytes without contacts, melanoma cells formed an attached layer along the JDE, several cells thick without noticeable dermal invasion (Figure 4b), nor breaks in collagen IV layer (Figure 4c). 

Zymography (Figure 4d), ELISA (Figure 4e) and western blot (Figure 4f) assays revealed that keratinocytes are the major source of proMMP-9 and that active MMP-9 was generated exclusively when Tspan8^+^ cells were surrounded with keratinocytes (Figure 4d–f). ProMMP-2, detected at low levels in the four types of culture, remained stable throughout the experiment (Figure 4d). Overall, our results indicate that interaction between Tspan8^+^ melanoma cells and neighboring keratinocytes are essential to drive MMP-9 activation, collagen IV dissolution, and subsequent dermal invasion.

### 2.4. Tspan8 Expression in Melanoma Cells Surrounded with Keratinocytes Promotes ProMMP-9 Activation by Increasing the Amount of Active MMP-3 and Decreasing TIMP-1 Levels

Since MMP-3 can activate in vitro proMMP-9 but not proMMP-2 [26,27] and because it is the most relevant activator of pro-MMP-9 in vivo [28,29], we wondered whether in our model proMMP-9 activation was MMP-3-dependent. Thus, total MMP-3 levels were measured in the media derived from our four culture models. MMP-3 highest amounts were observed when Tspan8^+^ melanoma cells were integrated with keratinocytes into the SR, peaking at day 21 (Figure 5a). MMP-3 was undetectable when Tspan8^+^ cells were cultured alone. MMP-3 activation status was examined by Western blot (Figure 5b) and we observed a 52 kDa band corresponding to the molecular weight of proMMP-3 in all types of cultures, except the culture with melanoma cells alone, indicating that proMMP-3 is generated by keratinocytes and not melanoma cells. However, the 28 kDa band representing the fully activated form of MMP-3 [30] was restricted to co-cultures where Tspan8^+^ cells were surrounded with keratinocytes (Figure 5b). Importantly, MMP-3 full activation is Tspan8-dependent as SR generated with non-invasive melanoma cells ectopically expressing Tspan8 acquired the property to produce a large amount of fully active MMP-3 (Figure 5c). Concordantly, that property observed in SR containing Tspan8^+^ melanoma cells was abrogated when Tspan8 expression was silenced (Figure 5c). 

It was emphasized that active MMP-3 becomes a potent activator of proMMP-9 in a tumor cell model only when its concentration exceeds that of TIMP-1 [28]. We thus wondered whether active MMP-9, exclusively observed when Tspan8^+^ melanoma cells were surrounded by keratinocytes, coincided with low levels of TIMP-1. Indeed, of our four culture models, we found that secreted TIMP-1 is at its highest level when Tspan8^+^ cells are alone and at its lowest when they are in direct contact with keratinocytes (Figure 5d,e). Overall, our data show that Tspan8^+^ melanoma cells surrounded by keratinocytes maintained lower TIMP-1 level when compared to melanoma cells juxtaposed without contacts with keratinocytes, thus favoring proMMP-9 activation by the fully active MMP-3, in a Tspan8-dependent manner.

### 2.5. Keratinocytes are the Main Source of ProMMP-9 and ProMMP-3 Whereas Tspan8+ Melanoma Cells Are the Primary Source of TIMP-1

We next evaluated the cellular source of MMP-9, its activator MMP-3 and its inhibitor TIMP-1. To this end, their respective transcript levels in keratinocytes alone (aK), Tspan8^+^ melanoma cells alone (bM), keratinocytes cocultured with melanoma cells without contact (cK and cM respectively), and melanoma cells that have penetrated the DEJ (dM) (Figure 6a) were measured by RT-QPCR. *MMP-9* and *MMP-3* were mainly expressed by keratinocytes whereas *TIMP-1* was mainly expressed by Tspan8^+^ melanoma cells (Figure 6b). The presence of keratinocytes, irrespective of contacts with melanoma cells, slightly augmented the *MMP-9* mRNA levels in keratinocytes and sorely decreased *TIMP-1* transcription in invasive melanoma cells. Surprisingly, *MMP-9* and *MMP-3* transcripts were increased in invading melanoma cells that have penetrated the DEJ, in comparison to melanoma cells still in contact with the DEJ. This was consistent with the immunodetection of MMP-9 and MMP-3 in melanoma cells located into the dermis, but not those localized in the epidermis (Figure 6c). This indicates that Tspan8^+^ melanoma cells, once invading the dermis, acquire the capability of expressing the precursor forms of MMP-9 and MMP-3, which were previously provided by the keratinocytes when situated in the epidermis. 

### 2.6. Antibody-Specific Blockade of Tspan8 Reduces ProMMP-9 Activation and Melanoma Invasion

We next examined whether a blocking monoclonal anti-Tspan8 antibody, previously shown to be effective in delaying the growth of human colon xenografts [31], could influence melanoma invasion. First, we showed that it allows efficient selective in vivo imaging of Tspan8^+^ human melanoma xenografts, demonstrating its high target specificity (Figure 7a). When this antibody was added to the culture medium of SR, Tspan8^+^ M#1 cells grow as clusters in the epidermis without deeply invading the dermis, whereas isotype-matching control Ab-treated SR invaded the dermis by day 10 and progressed deeper by day 20 (Figure 7b). The invasion score confirmed that Tspan8^+^ cells treated with Tspan8 mAb exhibited minimal invasion (mean score 1.83; *n* = 6) when compared to cells cultured with control mAb (mean score 0.67, *n* = 6) in a statistically significant manner (*p* = 0,01267; paired t-test; Figure 7b). These findings were extended to the SKMel28 cell line, broadly used for its ability to invade the dermis of SR [25], and revealed to be Tspan8^+^ [12]. As depicted in Figure 7c, SKMel28 cells in SR displayed vertically orientated clusters in the upper dermis when control mAb was added. In contrast, tumor nodules remained close to DEJ with less dermal invasion when treated with Tspan8 mAb (Figure 7c). Importantly, Tspan8-mAb treatment correlated with a strong reduction in MMP-9 activation in the SR integrating M#1 and SKMel28 cells (Figure 7d).

## 3. Discussion 

To date, little information is available regarding the epidermal microenvironment role in the proteolytic events involved in breaking the dermal-epidermal junction, a prerequisite for melanoma invasion. Here, we demonstrate, using a skin-reconstruct model that closely mimics the tumor microenvironment in vivo, that melanoma cells require the presence of neighboring keratinocytes within a fully differentiated epidermis to promote dermal invasion. This is in agreement with Eves et al. [32] and Van Kilsdonk et al. [33], who showed that melanoma cells invade the dermis only when integrated into the epidermis. However, the mechanisms of this process have not been explored. Our data reveal that Tspan8 drives mutual cooperation between melanoma cells and epidermal microenvironment to trigger the proMMP-9 activation process primarily produced by the keratinocytes, leading to collagen IV-containing DEJ proteolysis and dermal invasion. 

MMP-9 overexpression is traditionally associated with cancer aggressiveness and poor prognosis [34]. However, contradictory data have been reported in melanoma. Van den Oord et al. [35] found that MMP-9 was mostly expressed in primary lesions <1.6 mm, but not in metastases. Hofmann et al. [36] reported that several melanoma cell lines derived from metastases did not express MMP-9 at both mRNA and protein levels. Conversely, Simonetti et al. [37] report the highest MMP-9 levels in melanomas >2 mm thick. In line with this, MacDougall et al. [38] showed that MMP-9 was expressed in melanoma cell lines established from patient metastases but not from primary lesions. Our own data indicate that melanoma cells in the epidermis, even when presenting an invasive potential due to Tspan8, do not express proMMP-9. However, after crossing the DEJ, melanoma cells exhibit increased capabilities for proMMP-9 expression. This implies that MMP-9 expression by melanoma cells is acquired after dermal invasion and local dissemination. Thus, it appears that cutaneous environment exerts a powerful selective pressure for the emergence of cells with increasingly aggressive traits, probably the source of the well-recognized intra- and inter-heterogeneity of melanoma lesions. This might play a decisive role in the initiation of melanoma spreading. Indeed, in a spontaneous metastasis model, Hofmann et al. [39] noticed that a majority of melanoma cells expressed MMP-9 in lung metastases. Nevertheless, it is still unclear whether and how MMP-9 produced by melanoma cells, nearby host cells, or both, might be involved in late-stage melanoma. In mice, forced MMP-9 expression in melanoma cells enhanced lung colonization [40] which was reduced in MMP-9-deficient mice [41], indicating that MMP-9 produced by neoplastic and host cells might be equally important for the initiation of metastatic spreading. 

Several other tetraspanins, mainly CD9, CD81, CD82 and CD151 have been described to regulate proMMP-2 and/or proMMP-9 expression in cancer cell lines from liver [42], kidney [43], breast [44] and lung [45] carcinomas, fibrosarcomas [46] and melanomas [47,48]. However, functionally relevant MMP-2/-9 active forms were never detected. This is the first study reporting the role of a tetraspanin family member, Tspan8, in coordinating heterotypic crosstalk between cancer cells and surrounding epithelial cells to promote basement membrane proteolysis and stromal invasion through an MMP activation process.

MMP-9 is secreted as a latent pro-enzyme that requires activation in the extracellular space to achieve catalytic activity [49]. In cellular models, active MMP-3 is considered the most potent proMMP-9 activator [28,29]. Here, we found that the production of active MMP-9 and fully active MMP-3 were Tspan8-dependent, concomitant, and always correlated with collagen IV breakdown and dermal invasiveness. This is consistent with other reports linking MMP-3 to invasion and metastatic potential of melanoma cell lines and shorter disease-free survival [50,51]. Our data, together with the data available in the literature, reveal that Tspan8 expression in melanoma cells promotes the activation of keratinocyte-generated proMMP-3 in the stroma, which engages MMP-9 activation and DEJ proteolysis.

MMP-9 proteolytic activity is also tightly regulated extracellularly by its physiological inhibitor TIMP-1 [49,52]. We observed high levels of proteolytically active MMP-9 concurrently with very low levels of melanoma-derived TIMP-1 at the time of collagen IV dissolution, exclusively when melanoma cells were integrated into SR and expressed Tspan8 (Appendix A). These findings are consistent with prior data demonstrating that TIMP-1 overexpression in B16-F10 melanoma cells reduces their invasive capacity [53] and their metastatic potential [54]. Overall, our results strongly suggest that, in the epidermis, Tspan8^+^ melanoma cells cooperate with surrounding keratinocytes to promote dermal invasion by instigating MMP-3 activation and strongly decreasing TIMP-1 expression, with both events leading to a keratinocyte-originated MMP-9 activation process, and subsequent DEJ penetration. A striking finding is that aggressive cells capable of invading the DEJ to reach the dermis also gain the ability to express MMP-9 and its activator MMP-3 with reduced TIMP-1 levels. This is consistent with previous data conducted in mice, where the selection process for metastatic subclones favors those expressing MMP-9 [40] and those expressing Tspan8 (Figure 1). This profile should allow them to escape the local tissue control, and thus degrade the basal membranes encountered later throughout the metastatic cascade by themselves.

A major discovery in this work is that the ability/inability to cross DEJ is interconvertible and that the switch from one state to another can be accomplished both at the functional and molecular level by simply manipulating Tspan8 expression. Accordingly, a Tspan8-specific antibody efficiently targeting in vivo Tspan8^+^ melanoma xenografts was able to reduce MMP-9 activity, DEJ breakdown, and dermal invasion. Given that MMPs inhibitors are not highly selective and did not impede a single MMPs function [55,56], it is tempting to speculate that targeting Tspan8 with antibodies might represent an alternative means to specifically block MMP-9 activity, and thereby deeper melanoma invasion of the dermis, the earliest stage before metastatic spreading.

## 4. Materials and Methods 

### 4.1. Cell lines and Culture

SKMel28 (ATCC, Manassas, VA, USA) and M4Be [10] human melanoma cell lines were derived from lymph node metastasis. Non-metastatic and metastatic subpopulations were selected from immunosuppressed newborn rats that had been subcutaneously injected with M4Be (parental) cells, from lung metastases collected and grown in culture as described previously [57]. Stable clones of human melanoma cells were generated with shRNA-mediated silencing or ectopic overexpression of Tspan8 as described elsewhere [21]. Cells were cultured under standard conditions and tested as mycoplasma-free.

### 4.2. Matrigel Invasion Assay 

Invasion assays were performed in triplicates using BioCoat Matrigel invasion chambers (BD Biosciences) as previously described [21]. Briefly, the cells that migrated to the lower surface of the filter were fixed, stained with DAPI, imaged using an Axiovert 200 (Carl Zeiss Inc., Jena, Germany) equipped with a CoolSNAP HQ camera (Roper Scientific, Lisses, France) and MetaMorph software (MDS Analytical Technologies, Sunnyvale, CA, USA) and then counted on the entire filter using NIH Image J software. 

### 4.3. Invasion Assay in Human Skin Reconstructs 

Adult human keratinocytes (4 × 10^5^ cells), mixed or not with human melanoma cells (5.820 cells) at a melanoma/keratinocyte ratio of 1:80, were seeded into a stainless-steel ring deposited on the surface of human dead de-epidermized dermis (DED) squares as previously described [24]. In some experiments, the same respective number of melanoma cells and keratinocytes were seeded alone onto the surface of DED. After 9, 15 and 21 days of incubation at an air-liquid interface, the specimens were collected and embedded in paraffin for hematoxylin and eosin staining or embedded in Tissue-Tek (Miles Inc., Elkhart, IN, USA) for further immunohistochemical staining of type IV collagen (clone CIV 22; Dako, Carpinteria, CA, USA) as described [24]. Four-micrometer vertical sections cut at different levels were subjected to histological and staining evaluation. To test the effect of anti-human Tspan8 antibody, SR were cultured in its constant presence (TS29 clone, 15 µg/mL; [31]) or an isotype-matching control antibody. Dermal invasion was evaluated by a scoring system of 0–2 in a blinded manner: 0 indicated no melanoma cells present in the dermis, 1 invasive melanoma cells were located under JDE, and 2 melanoma cells were observed deep into the dermis. All experiments were done as sixtiplates and were repeated twice for each condition.

### 4.4. Preparation of Serum-Free Culture Medium

Culture fluids from skin composites were harvested on days 10, 15 and 21. Two days before collection, SR were extensively washed and cultured in serum-free medium. The collected serum-free culture media was centrifuged to remove cellular debris, and concentrated 10-fold in a Centricon ultrafiltration apparatus, containing a polysulfone membrane with an exclusion limit of Mr 10.000 (Millipore, Molshein, France). Protein concentrations were measured by the Bradford method using a commercial kit (Bio-Rad Laboratories, Paris, France). Aliquots with equivalent protein contents were subjected to gelatin zymography, Western blotting, and ELISA assays.

### 4.5. Gelatin Zymography

The activity of electrophoretically separated gelatinolytic enzymes in the serum-free culture media was analyzed as described previously [58]. 

### 4.6. Western Blot Analysis

Western blotting was performed as previously described [12]. Antibodies against MMP-9 (polyclonal antibody, Dako, Trappes. France), MMP-3 (clone 552A4, Oncogene research Product, Boston, Mass) and TIMP-1 (clone 7-6C1; Oncogene Research Products) were used. 

Tspan8 was detected using a mouse monoclonal anti-Tspan8 antibody (TS29 clone [12,18,19,20,21]. Western blot quantifications were performed using ImageJ software. At least three independent biological replicates were performed. 

### 4.7. Measurement of MMP-9, TIMP-1 and MMP-3

Serum-free culture medium was screened for pro and active MMP-9, total MMP-3 and total TIMP-1 using the commercially available ELISA kits (Amersham Pharmacia Biotech, Saclay, France), following procedures recommended by the manufacturer. All experiments were performed in triplicate from six separate experiments and the results were expressed as ng/mg of total proteins ± SD. 

### 4.8. Real-Time RT-QPCR

Total RNA was extracted using the RNAeasy mini-kit (Qiagen, Germantown, MD, USA), reverse-transcribed into cDNA by PrimeScript™ RT reagent Kit (TaKaRa, Shiga, Japan) and analysed by real-time QPCR using SYBR® Premix ExTaq™II (TaKaRa, Shiga, Japan) on a Mx3000P real-time PCR system (Stratagene, Santa Clara, CA, USA) as described [21]. Results were obtained from at least three independent experiments and normalized to the 18 S rRNA expression level. The primers used are as follows: 18S-F: 5′-CGATGCGGCGGCGTTATT-3′; 18S-R: 5′-CCTGGTCTGTCTCATCCTCCC-3′; TSPAN8-F: 5’-TTGCTTCTGATCCTG CTCCT-3’; TSPAN8-R: 5′-AGGGCCTGCAGGTTCACACCAC-3′; MMP-9-F: 5′-CACTGTCCACCCCTCAGAGC-3′; MMP-9-R: 5′-GCCACTTGTCGGCGATAAGG-3′; MMP-3-F: 5′-GGAAGCTGGACTCCGACACTC-3′; MMP-3-R: 5′-TGGTGTATAATTCACAAT CCTGTATGTAA-3′; TIMP-1-F: 5′-GACGGCCTTCTGCAATTCC-3′; TIMP-1R: 5′-GTATAAGGTGGTCTGGTTGACTTCTG-3′.

### 4.9. Flow Cytometric Analysis 

Cell surface labeling was performed as previously described [12]. Data were collected on a FACSCanto II (BD Biosciences, San Jose, CA, USA) and analyzed using FlowJo Software (Treestar, Ashland, OR, USA).

### 4.10. Animal Studies

NMR1 Foxn1nu/Foxn1nu female mice (Janvier Labs; Le Genest-Saint-Isle, France) were maintained and used in accordance with the 2010/63/UE directive after approval by the institutional review board C2E2A and the French MESR ministry. Mice were injected subcutaneously with 1.106 Tspan8+ or Tspan8- melanoma cells in their left or in right shoulder, respectively. Radiolabeling of Tspan8 mAB with 111Indium was performed as previously described [59]. The mice were imaged at each timepoint using a γ-camera (γ IMAGER, BIOSPACE Inc., Urbandale, IA 50322, USA) under gaseous anesthesia (Isoflurane, Iso-Vet® 1000 mg/g). Removed tumors were weighted and counted using a Wallac 1480 automated calibrated γ-counter (Perkin-Elmer, Waltham, MA, USA).

### 4.11. Statistical Analysis

Statistical significance was calculated by a two-tailed Student’s *t*-test for unpaired samples. Mean differences were considered to be significant when *p* < 0.05. 

## 5. Conclusions

In summary, we report the novel finding that within a human-differentiated epidermis, Tspan8 expression in melanoma cells cooperate with surrounding keratinocytes to promote dermal invasion by instigating keratinocyte-produced MMP-3 activation and decreasing melanoma-derived TIMP-1 levels, leading to keratinocyte-originated MMP-9 activation process, and subsequent DEJ-collagen IV degradation. Furthermore, an anti-Tspan8 monoclonal antibody specifically targeting Tspan8^+^ melanoma xenografts in vivo significantly reduces dermal invasion by strongly impairing proMMP-9 activation process and collagen IV breakdown. 

This study is the first to provide evidence for the pro-invasive role of Tspan8 in a cell non-autonomous manner, a mechanism never reported for a tetraspanin family member. This work has important implications since the direct inhibition of MMPs proved disappointing in clinical trials, and therefore targeting Tspan8 might represent a novel alternative and efficient strategy to impede MMP-9 proteolytic activity and greatly reduce metastasis risks. 

## Figures and Tables

**Figure 1 cancers-12-01297-f001:**
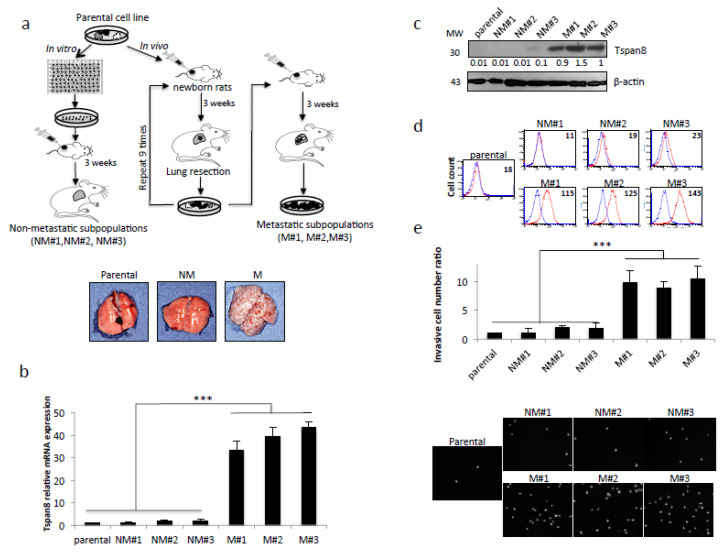
Generation of potent metastatic cell subpopulations expressing the metastatic-associated Tspan8 protein. (**a**) Schematic diagram of the experimental procedure used to sequentially select in an immunosuppressed new-born rat model cell subpopulations with progressively higher metastatic ability from a poorly metastatic melanoma cell line. Lower panel, representative photographs of the rat lungs. (**b**) The parental human M4Be cell line and its derived non metastatic (NM#1-3) and metastatic (M#1-3) subpopulations were examined for *TSPAN8* mRNA levels by QPCR. Expression normalized to GAPDH represented a fold change of control sample (*n* = 3; ± SD); (**c**) Western blot analysis of Tspan8 expression with β-Actin as loading control and reference for quantification (one representative experiment of three), uncropped western blots figures in Appendix A; (**d**) Tspan8 cell surface expression by flow cytometry analysis. In red, the specific staining and in blue the isotype-matched control antibody (one representative experiment of three). Numbers indicate Mean Fluorescence Intensity (MFI). (**e**) Matrigel invasion assay using transwell chambers. The total number of invasive cells was integrally counted by scanning microscopy and normalized to the value from control parental cell line (*n* = 3; ± SEM). Representative visual fields are illustrated beneath. ****p* < 0.001.

**Figure 2 cancers-12-01297-f002:**
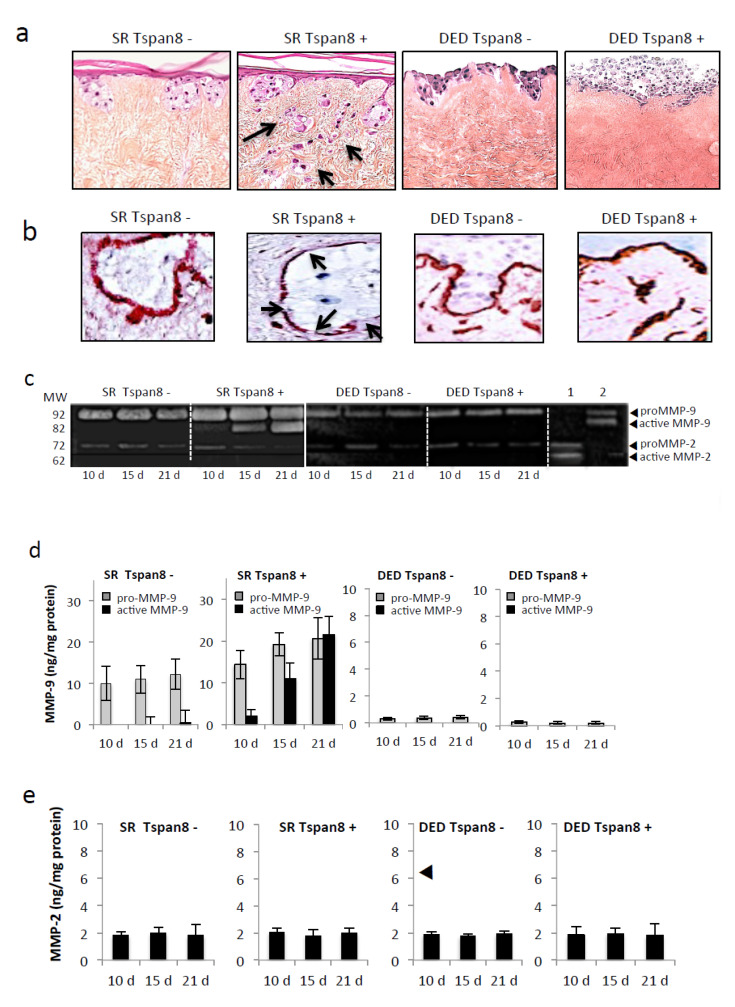
Tspan8-expressing melanoma cells efficiently invade the dermis in human skin reconstructs. Melanoma cells from NM#1 (Tspan8-) and M#1 (Tspan8+) subpopulations were cultured with human keratinocytes (SR) or alone (DED) on acellular dermis. (**a**) Representative photomicrographs of hematoxylin and eosin (H&E)-stained 21-day skin composites (scale bars: 100 μm). Arrows indicate melanoma cells colonizing the dermis (**b**) Representative IHC-staining of collagen IV. Arrows denote collagen IV layer disruptions. (**c**) MMP-9 and MMP-2 activity in gelatin zymography of culture medium from skin composites collected on day 10, 15 and 21. Lane 1, purified MMP-2 standard; lane 2, purified MMP-9 standard both activated with 4-aminophenylmercuric acetate. (**d**,**e**) ELISA quantification of secreted protein levels of proMMP-9, active MMP-9 (d) and MMP-2 (e) (ng/ug total protein) into the composite media. Bars represent the mean ± SD of three separate experiments with 3 ELISA evaluations for each of the 3 independent experiments (*n* = 9).

**Figure 3 cancers-12-01297-f003:**
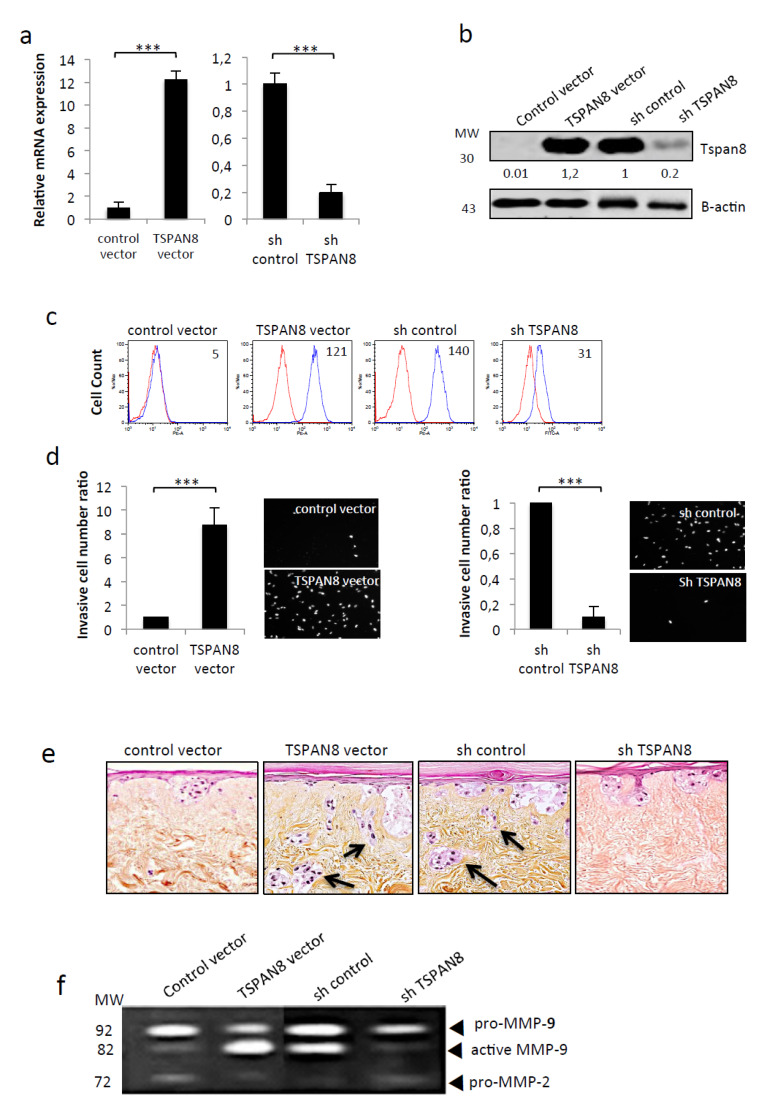
Tspan8 expression in melanoma triggers dermal invasion, concomitantly to MMP-9 activation and collagen IV proteolysis. Non-metastatic stable clones ectopically-expressing Tspan8 (TSPAN8 vector) or not (control vector) Tspan8 (left panel) and metastatic stable clones silenced (shTSPAN8) or not (shcontrol) for Tspan8 (right panel) were subjected to (**a**) QPCR analysis of *TSPAN8* transcripts levels (*n* = 3; mean ± s.d.). (**b**) Western blot analysis of Tspan8 protein levels with β-Actin as loading control. The band intensities were normalized to actin signal (representative experiment of three), uncropped western blots figures in Appendix A (**c**) Flow cytometry analysis of cell surface Tspan8 expression (representative experiment of three). (**d**) Matrigel cell invasion assay: invading cells were DAPI-stained (right panel) and quantified (left panel). Data are means ± SD with *n* = 3 (*** *p* < 0.001). (**e**) Cells were incorporated into the epidermis of skin reconstructs as described in Materials and Methods. Representative hematoxylin and eosin-stained skin reconstruct were shown. Arrows denote melanoma cells located into the dermis. (**f**) Serum-free conditioned media collected from SR were analyzed by gelatin zymography at 21 days (representative zymogram of 3 independent experiments). Molecular weight (MW) markers are indicated in kDa.

**Figure 4 cancers-12-01297-f004:**
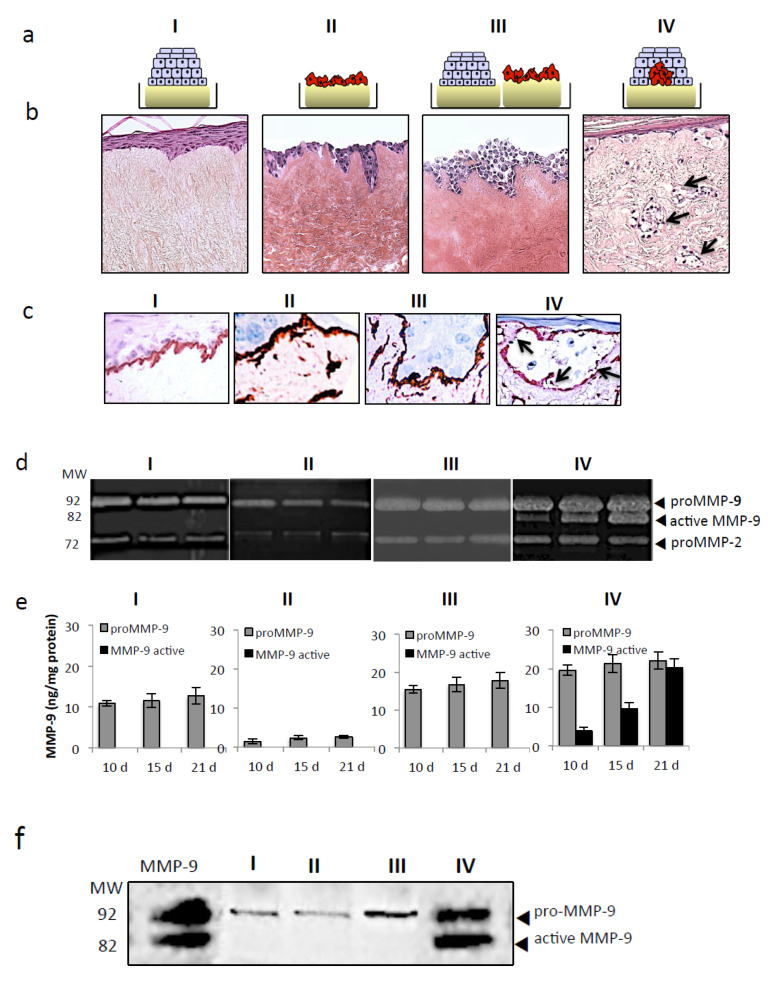
Tspan8-dependent dermal invasion coincides with MMP-9 activity and local dissolution of collagen IV and requires surrounding keratinocytes. (**a**) Schematic drawings of the four different culture conditions. I: SR containing no melanoma cells; II: Tspan8-expressing cells seeded alone on DED (DeEpidermised Dermis); III: SR without melanoma cells juxtaposed with Tspan8+ cells seeded alone on DED; IV: SR containing Tspan8+ cells in contact with keratinocytes. (**b**) Representative H&E staining of skin composites sections. Arrows indicate melanoma cells infiltrating the dermis. (**c**) type IV collagen staining on sections from the four culture conditions described in (**a**). Arrowhead pointed to collagen IV destruction. (**d**–**f**) Serum-free media from the four culture conditions collected at day 10, 15 and 21 were analyzed for the expression levels of proMMP-9 and active MMP-9 using gelatin zymography (**d**), ELISA (**e**) and western blot (**f**).

**Figure 5 cancers-12-01297-f005:**
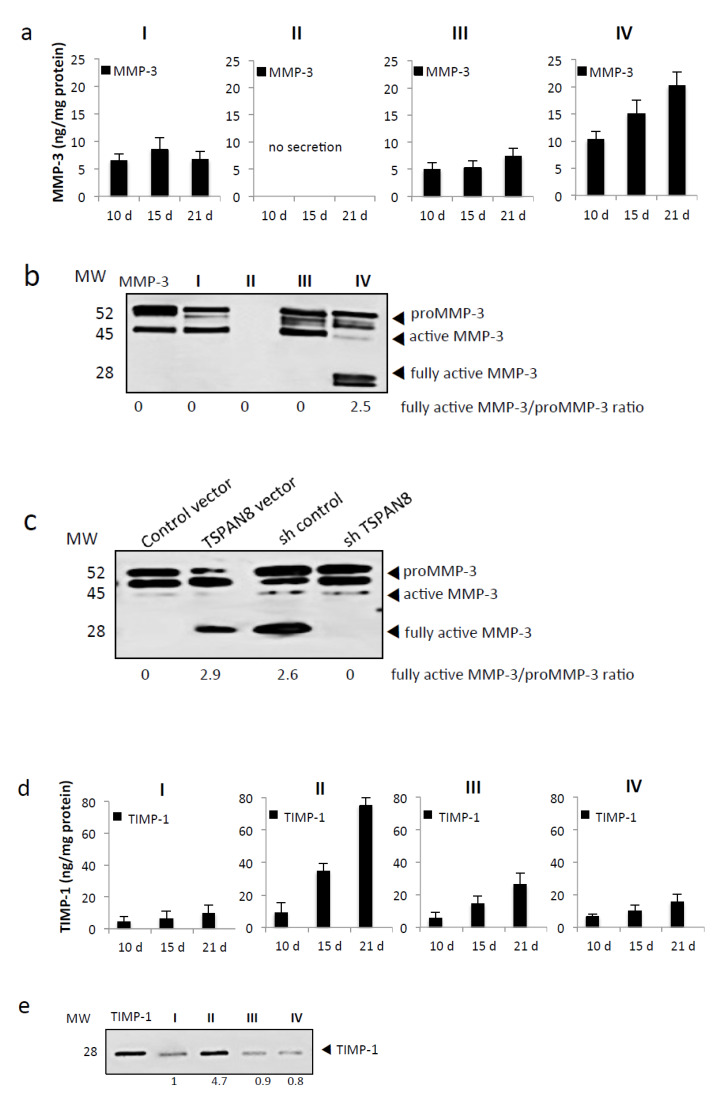
Tspan8-mediated melanoma cell invasion coincides with MMP-3 activation concomitantly with low TIMP-1. (**a**,**b**) The protein levels of total MMP-3 released in the supernatants from each culture model (I, II, II, IV described in Figure 4) were assessed at day 10, 15 and 21 by ELISA (**a**) and western blot (**b**), uncropped western blots figures in Appendix A. (**c**) Immunoblot analysis of MMP-3 in serum-free media harvested from SR integrating non-metastatic NM#1 melanoma cells ectopically expressing Tspan8 (TSPAN8 vector) and their control (control vector) or metastatic M#1 melanoma cells silenced (shTSPAN8) or not (shcontrol) for Tspan8. Equal amounts of total protein were loaded. The intensity value ratio of fully active MMP-3/proMMP-3 was annoted beneath the blot (**d**,**e**). Serum-free media from the 4 culture models collected at day 10, 15 and 21 were analyzed for TIMP-1 content by ELISA (**d**) and western blot (**e**). ELISA results are represented as the mean ± SEM from three independent experiments, each measured in duplicate.

**Figure 6 cancers-12-01297-f006:**
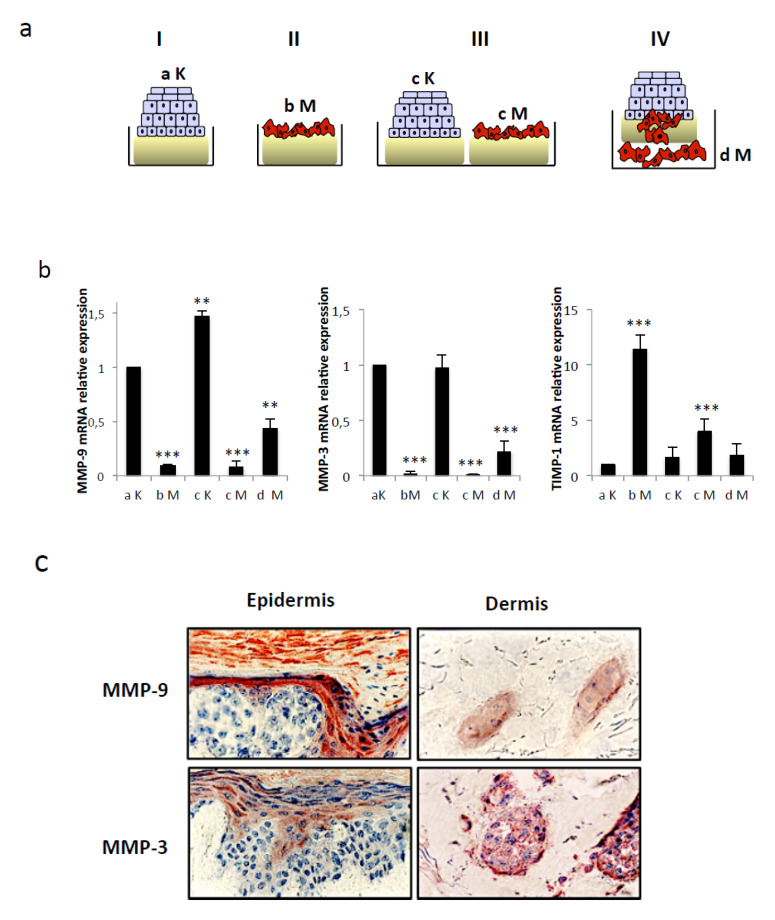
Keratinocytes are the main source of MMP-3 and MMP-9 in the epidermis but melanoma cells gain the ability to express both proteins after DEJ crossing. (**a**) Total RNA has been isolated from keratinocytes and Tspan8+ melanoma cells at day 20 from the four different schematized culture conditions. aK: keratinocytes from culture I; bM: Tspan8+ melanoma cells from culture II; cK and cM: keratinocytes and Tspan8+ melanoma cells from culture III respectively; dM: invading melanoma cells from culture IV. (**b**) QPCR analysis of *MMP-9*, *MMP-3*, and *TIMP-1* transcript expression levels of aK, bM, cK, cM and dM (*n* =3; ± SD). (**c**) Representative pictures of immunohistochemical staining of MMP-9 and MMP-3 in the epidermis and dermis of SR integrating Tspan8+ melanoma cells (condition IV) at day 20. ** *p* < 0.01, *** *p* < 0.001.

**Figure 7 cancers-12-01297-f007:**
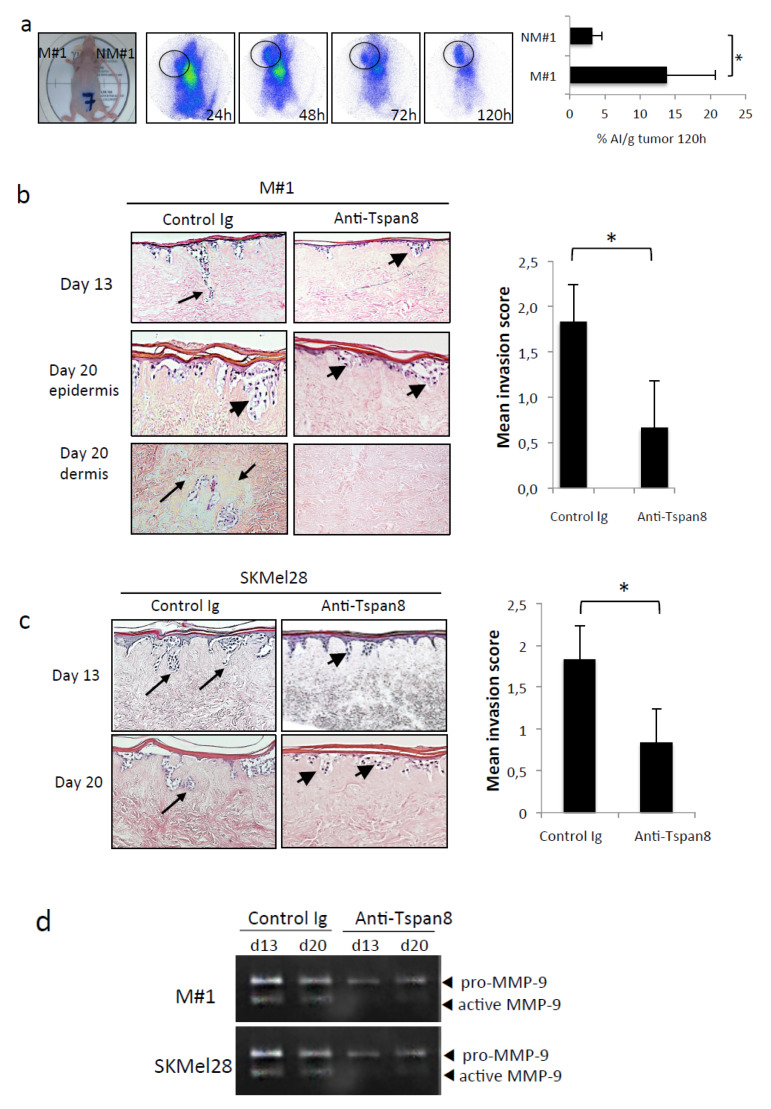
Anti-Tspan8 mAb efficiently targets Tspan8-positive melanoma cells in vivo and reduces MMP-9 activation and dermal invasion. (**a**) Mice with Tspan8+ (left side) and Tspan8- xenografts (right side) were injected (i.v.) with 3.7 MBq of [111In] DOTA-mAb and imaged with a γ-camera at 24 h, 48 h, 72 h and 120 h post-injection. Whole body SPECT/CT images of mice demonstrate specific accumulation of [111In] DOTA-mAb in Tspan8+ tumors (surrounded) but significantly lower in Tspan8- tumors. Tumors were collected 120 h after injection and the radioactivity was measured by γ-counting of each sample. The graph represents the % of injected activity per gram of tissue (%IA/g, *n* = 4). (**b**) Representative H&E staining of SR with metastatic M#1 cells treated with control IgG or 0.5 µM Ts29 at 13 and 20 days. The graph depicts the invasion scores (see Materials and Methods). (**c**) Representative H&E staining of SR with SKMel28 cells treated with control IgG or 15 µg/mL Ts29 at 15 and 21 days. Arrow heads: melanoma cell clusters close to DEJ; arrows, melanoma cells invading the dermis (Scale bars: 10 μm). Data representative of 6 independent experiments. Graph depicts results of invasion score analysis. (**d**) Serum-free conditioned media collected from SR integrating M#1 and SKMel-28 cells were analyzed by gelatin zymography at 21 days (representative zymogram of 3 independent experiments). * *p* < 0.05.

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
