# Peer review of "Tspan8 Drives Melanoma Dermal Invasion by Promoting ProMMP-9 Activation and Basement Membrane Proteolysis in a Keratinocyte-Dependent Manner"

_cancers, 2020, doi:10.3390/cancers12051297_

Round 1

Reviewer 1 Report

The present research is interesting and original, materials and methods are clearly explained and the experiments are well conducted. 

Along with minor corrections to English language, I suggest some comments:

  • Introduction (Lines 45-53) should be implemented with more precise information on known prognostic factors of melanoma, citing also the new staging system (AJCC VIII edition), and providing information on different subtypes of melanoma (at least differentiating cutaneous, mucosal and uveal melanomas).
  • Lines 54-56: even if the role of the immune system might sound off-topic in the present research, I would suggest to cite it when addressing the pathological genotypes and phenotypes of melanoma.
  • Lines 364-367: Authors should not suggest targeting Tspan as a novel potential therapeutic strategy. To date, there are still limited data in this field, without clinical trials ongoing. Of course the molecular mechanisms presented here are interesting and should be further investigated, however to date this is far from offering a potential change in the therapeutic strategy of melanoma.
  • Pathogenesis of melanoma involves several important signaling pathways, along with the fundamental role of the host’s immune system, and there are several novel drugs that have demonstrated to successfully act on these targets. Maybe the role of Tspan should be investigated as a possible co-target, to improve therapeutic outcomes and/or prevent metastatic spreading and disease progression.

Author Response

Authors’ Response to the Review Comments

Manuscript ID: Cancers-792028

Dear Ms. Simon You and Reviewers,      

We are pleased to have the opportunity to submit a revised version of our manuscript. We sincerely appreciate all insightful reviewer’s comments, which helped us to improve the quality of the article. Please, find below a point-by-point reply to the reviewer’s comments. All changes made from the previous version have been introduced to the manuscript in blue in the presently resubmitted version. We have also added five publications in the introduction; the references numbers have been modified accordingly all along the manuscript.

We hope that this revised version of our manuscript will satisfy the Reviewers and will be suitable for publication in Cancers.

On behalf of all the co-authors

Sincerely yours,

Drs Odile Berthier-Vergnes and Ingrid Masse

Response to Comments from Reviewer # 1

The present research is interesting and original, materials and methods are clearly explained and the experiments are well conducted. 

Along with minor corrections to English language, I suggest some comments:

  • Introduction (Lines 45-53) should be implemented with more precise information on known prognostic factors of melanoma, citing also the new staging system (AJCC VIII edition), and providing information on different subtypes of melanoma (at least differentiating cutaneous, mucosal and uveal melanomas).

The Reviewer #1 is right to point out that malignant melanoma arises from different body compartments, the major subtypes including cutaneous, acral, mucosal and uveal melanomas. In the present study, we only focused on cutaneous melanoma. Therefore, this point has been clarified by adding « cutaneous melanoma« in the introduction (line 45 and 59 of the revised manuscript).

Moreover, as requested, we have also clarified the prognostic factors of cutaneous melanoma according to the 8th Edition of the AJCC staging system by including the following new paragraph supplemented with two novel references in the Introduction of the revised manuscript (lines 49-54).

« To date, Breslow thickness remains the most powerful prognostic factor, as long as metastases are not present at the time of diagnosis. The most recent AJCC 8th guidelines introduced mitotic rate as an additional criterion for thinner melanomas, the presence of > 1 mitosis/mm² predicts a poorer outcome [3]. Moreover, the ulceration status used for sub-classification of thin melanomas [3], emerges as another important histological factor predicting survival [4]. However, such histological features define prognostic groups but not individual patient risk ».

  1. Gershenwald, J.E.; Scolyer, R.A.; Hess, K.R.; Sondak, V.K.; Long, G.V.; Ross, M.I.; Lazar, A.J.; Faries, M.B.; Kirkwood, J.M.; McArthur, G.A.; et al. Melanoma staging: Evidence-based changes in the American Joint Committee on Cancer eighth edition cancer staging manual. C.A. Cancer J. Clin. 2017, 67, 472-492.
  2. Roncati, L.; Piscioli, F. AJCC 8th Edition (2017) versus AJCC 7th Edition (2010) in thin melanoma staging. Neoplasma. 2018, 65, 651-655.

  • Lines 54-56: even if the role of the immune system might sound off-topic in the present research, I would suggest to cite it when addressing the pathological genotypes and phenotypes of melanoma.

In accordance with the wishes of the Reviewer #1, we included a new paragraph in the introduction (lines 59-65) supplemented with 3 novel references in the revised version.

« Cutaneous melanomas are composed of genotypically and phenotypically distinct subpopulations, dynamically regulated by the selective pressure imposed from host tumor microenvironment and host immune system [7]. This tumor heterogeneity contributes largely to their strong resistance to standard, targeted and immune therapies [8]. Indeed, it appears that cancer/immune cell interactions are informative of resistance to immunotherapy whereas cancer/stromal cell interactions are informative of MAPK inhibitors resistance [9]. 

  1. Flemming, A. Tumour heterogeneity determines immune response. Nat. Rev. Immunol. 2019, 19, 662–663.
  2. Arozarena, I.; Wellbrock, C. Phenotype plasticity as enabler of melanoma progression and therapy resistance. Nat Rev Cancer. 2019, 19, 377-391.
  3. Fattore, L.; Ruggiero, C.F.; Liguoro D.; Mancini, R.; Cilibert, G. Single cell analysis to dissect molecular heterogeneity and disease evolution in metastatic melanoma. Cell Death. Dis. 2019, 10, 827-833.

  • Lines 364-367: Authors should not suggest targeting Tspan as a novel potential therapeutic strategy. To date, there are still limited data in this field, without clinical trials ongoing. Of course the molecular mechanisms presented here are interesting and should be further investigated, however to date this is far from offering a potential change in the therapeutic strategy of melanoma.

To date, CD37 is the only targeted tetraspanin that has moved forward into the clinic to treat B cell lymphoid malignancies (de Winde CM et al, Trends Cancer 3(6):442, 2017) and recently reported to improve the overall survival (Payandeh, Z. et al, Biotechnol Lett 40, 1459; 2018 ; Witkowska M et al, Expert Opin Investig Drugs. 2018;27(2):171). I agree with the reviewer that our work only demonstrates that Tspan8-blocking antibodies decreases invasion of melanoma cells into dermis in a 3D reconstructed skin model through inhibition of MMP-9 activation process. Consequently, we have discussed these findings and just wanted to conclude that targeting Tspan8 with antibodies might represent an alternative means to inhibit specifically MMP-9 proteolytic activity, given that MMPs inhibitors are not highly selective and did not impede a single MMPs function.

In agreement with the Referee’s comments, we rephrased the end of the discussion with a novel paragraph (lines 378-381) in the revised version.

  • Pathogenesis of melanoma involves several important signaling pathways, along with the fundamental role of the host’s immune system, and there are several novel drugs that have demonstrated to successfully act on these targets. Maybe the role of Tspan should be investigated as a possible co-target, to improve therapeutic outcomes and/or prevent metastatic spreading and disease progression.

We thank the reviewer for this relevant suggestion. Actually, in the lab I recently joined, we previously demonstrated that the transcription factors regulating epithelial-to-mesenchymal transition (EMT-TFs) are crucial in melanoma progression (Caramel et al, Cancer Cell, 2013) and that ZEB1 in particular is implicated in resistance to MAPK inhibitors (Richard, EMBO Mol Med, 2016). In the line of the fundamental role of the host’s immune system and the use of targeting of immune pathways for therapies, we are currently studying the role of EMT-TFs in the regulation of the immune microenvironnement along melanoma progression and the potential interaction with Tspan8 in this context.

Reviewer 2 Report

The authors present a very interesting study about the role of tetraspanin-8 in the communication of melanoma cells with keratinocytes during the beginning of melanoma cell invasion and dissemination. Their findings that keratinocytes generate activated MMPs that support melanoma cell invasion in a TSPAN8 dependent manner is a novelty.

The authors should consider the following points:

  1. The official gene nomenclature should be used (HUGO). Human gene symbols generally are italicised, with all letters in uppercase (e.g., SHH, for sonic hedgehog). Italics are not necessary in gene catalogs. Protein designations are the same as the gene symbol except that they are not italicised.

  2. Please check the western blots in Figure 1c and the supplementary file. The blots in the supplement show different sample numbers so please make sure that the right bands were chosen. Could you do the protein detection on the same blot to avoid that problem?

  3. An additional melanoma marker (Melan-A, SOX10 etc) should be detected by western blot or IHC for the non-metastatic and metastatic cell lines from the Rat melanoma model. This would prove that they are really melanoma. Are these cell lines BRaf or Nras mutated? Could you detect the key driver in the different cell lines?

Author Response

Authors’ Response to the Review Comments

Manuscript ID: Cancers-792028

Dear Ms. Simon You and Reviewers,    

We are pleased to have the opportunity to submit a revised version of our manuscript. We sincerely appreciate all insightful reviewer’s comments, which helped us to improve the quality of the article. Please, find below a point-by-point reply to the reviewer’s comments. All changes made from the previous version have been introduced to the manuscript in blue in the presently resubmitted version. We have also added five publications in the introduction; the references numbers have been modified accordingly all along the manuscript.

We hope that this revised version of our manuscript will satisfy the Reviewers and will be suitable for publication in Cancers.

On behalf of all the co-authors

Sincerely yours,

Drs Odile Berthier-Vergnes and Ingrid Masse

Response to Comments from Reviewer # 2

The authors present a very interesting study about the role of tetraspanin-8 in the communication of melanoma cells with keratinocytes during the beginning of melanoma cell invasion and dissemination. Their findings that keratinocytes generate activated MMPs that support melanoma cell invasion in a TSPAN8 dependent manner is a novelty.

The authors should consider the following points:

1-The official gene nomenclature should be used (HUGO). Human gene symbols generally are italicised, with all letters in uppercase (e.g., SHH, for sonic hedgehog). Italics are not necessary in gene catalogs. Protein designations are the same as the gene symbol except that they are not italicised.

As requested, the human genes TSPAN8, MMP-3, MMP-9, TIMP-1 have been corrected using capital and italic letters all along the revised version of the manuscript.

2-Please check the western blots in Figure 1c and the supplementary file. The blots in the supplement show different sample numbers so please make sure that the right bands were chosen. Could you do the protein detection on the same blot to avoid that problem?

We thank the Reviewer #2 for this clarification. We have therefore checked the correspondence between the lanes of the 2 membranes: the right bands have been chosen.

For the detection of Tspan8, the proteins are run on a non-reducing gel (also valid for other tetraspanins) whereas for actin, we need reducing conditions. Indeed, the disulfide bonds in the large extracellular loop (LEL) of tetraspanins are important for the 3D structure and the antibodies directed against tetraspanins recognized these conformational epitopes. This is why we cannot detect Tspan8 and actin on the same blot.

3-An additional melanoma marker (Melan-A, SOX10 etc) should be detected by western blot or IHC for the non-metastatic and metastatic cell lines from the Rat melanoma model. This would prove that they are really melanoma. Are these cell lines BRaf or Nras mutated? Could you detect the key driver in the different cell lines?

We thank the Reviewer #2 for his comments and would like to address this concern within a few points. We previously analyzed the expression of the two commonly used protein markers HMB-45 and Melan-A in the in vivo selected non-metastatic and metastatic melanoma cell clones. Surprisingly, we found that metastatic cell clones cannot be identifiable using any of the markers, as also noticed by Haridas et al. (Scientific Reports, 6:24569; 2016) in other metastatic melanoma cell lines. In agreement with this, Viray et al (Arch Pathol Lab Med. 2013;137(8):1063-73) reported a great heterogeneity of HMB-45, Melan A and S100 staining in melanoma cells from a large series of primary melanomas.

Therefore, to prove that the in vivo selected clones are indeed melanoma cells, their karyotypes were analyzed, given that we have used a heterotransplantation rat model of human tumor cells. The clones all exhibit an aneuploidy karyotype with banding characteristics of human chromosome, proving that they are indeed human and therefore, melanoma cells (Bailly M et al. Melanoma Res, 1993,3:51-61). Concerning SOX10, we did not carry out experiments and we regret to be unable to provide such data.

The parental melanoma cell line as well as its in vivo-derived invasive and non invasive cells used in the present study were BRAFV600E mutated (data not shown). Moreover, we previously reported that invasive melanoma cells treated by the BRAF inhibitor PLX4032 (vemurafenib) exhibit a strong reduction of Tspan8 protein expression level, and that silencing of Tspan8 by siRNA induces a decrease in the number of invasive melanoma cells resistant to PLX4032 (Agaesse et al, Oncogene. 2017;36(4):446-457), suggesting that Tspan8 could represent a putative target in BRAF-resistant melanomas.

Reviewer 3 Report

The manuscript entitled: “Tspan8 drives melanoma dermal invasion by promoting proMMP-9 activation and basement membrane proteolysis in a keratinocyte-dependent manner” by Kharbili et al., describes how the interaction between metastatic melanoma cells (Tspan8+) and keratinocytes promotes the dermal-epidermal junction degradation and dermal invasion. Here the authors described a role for Tspan8 in tumor invasion. They also proved that Tspan8 can activate MMP-9 in the presence of keratinocytes. The work is novel, the results are clear and the conclusions are supported by the results. However, few comments and question need to be addressed by the author before recommend this manuscript for publication in Cancers.

In figure 1 the authors showed the generation of highly metastatic melanoma cell lines that express Tspan8. They evaluated the ability to invade matrigel. However, the description of the experiment refers to a self-citation where no matrigel experiment was described or performed. The authors need to provide the correct reference or a detailed description of the experiment.

2.2 Tspan8 expression in melanoma cells promotes proMMP-9 activation, collagen IV degradation and DEJ crossing.

In this section the authors claim that Tspan8 is required to activate proMMP-9 and the subsequence collagen IV degradation and DEJ crossing. Here they mention “Evidence of collagen IV dissolution, the major DEJ component, were observed exclusively when Tspan8+ cells were used (Figure 2b)”, and this is only true in the presence of keratinocytes. Tspan8+ cells without co-culture with keratinocytes also express MMP-9 but they are unable to activate it, indicating that keratinocytes are required not only to provide proMMP-9.  The author should explain this in more detail.

2.4 Tspan8 expression in melanoma cells surrounded with keratinocytes promotes proMMP-9 activation by increasing the amount of active MMP-3 and decreasing TIMP-1 levels.

Here the authors correlated the fully activation of MMP3 which is produced by keratinocytes, with co-culture with Tspan8+ melanoma cells. However, the claim that Tspan8 decreases the levels of TIMP-1 is not supported by the results. Since the co-culture is in a 1:80 ratio (melanoma cell:keratinocyte), how the authors differentiate between downregulation of TIMP-1 and dilution with low expressing keratinocyte? Additional data need to be provided to support their claim.

Overall the manuscript was well written and I think it will be of interest for the scientific community.

Author Response

Authors’ Response to the Review Comments

Manuscript ID: Cancers-792028

Dear Ms. Simon You and Reviewers,    

We are pleased to have the opportunity to submit a revised version of our manuscript. We sincerely appreciate all insightful reviewer’s comments, which helped us to improve the quality of the article. Please, find below a point-by-point reply to the reviewer’s comments. All changes made from the previous version have been introduced to the manuscript in blue in the presently resubmitted version. We have also added five publications in the introduction; the references numbers have been modified accordingly all along the manuscript.

We hope that this revised version of our manuscript will satisfy the Reviewers and will be suitable for publication in Cancers.

On behalf of all the co-authors

Sincerely yours,

Drs Odile Berthier-Vergnes and Ingrid Masse

Response to Comments from Reviewer # 3

The manuscript entitled: “Tspan8 drives melanoma dermal invasion by promoting proMMP-9 activation and basement membrane proteolysis in a keratinocyte-dependent manner” by Kharbili et al., describes how the interaction between metastatic melanoma cells (Tspan8+) and keratinocytes promotes the dermal-epidermal junction degradation and dermal invasion. Here the authors described a role for Tspan8 in tumor invasion. They also proved that Tspan8 can activate MMP-9 in the presence of keratinocytes. The work is novel, the results are clear and the conclusions are supported by the results. However, few comments and question need to be addressed by the author before recommend this manuscript for publication in Cancers.

- In figure 1 the authors showed the generation of highly metastatic melanoma cell lines that express Tspan8. They evaluated the ability to invade matrigel. However, the description of the experiment refers to a self-citation where no matrigel experiment was described or performed. The authors need to provide the correct reference or a detailed description of the experiment.

We apologize for this mistake and have corrected this by replacing the reference 17 by 21 (El Kharbili et al, Oncogene. 2019, 38, 3781-3793) in Materials and Methods, section “matrigel” of the revised version (line 394).

2.2 Tspan8 expression in melanoma cells promotes proMMP-9 activation, collagen IV degradation and DEJ crossing.

In this section the authors claim that Tspan8 is required to activate proMMP-9 and the subsequence collagen IV degradation and DEJ crossing. Here they mention “Evidence of collagen IV dissolution, the major DEJ component, were observed exclusively when Tspan8+ cells were used (Figure 2b)”, and this is only true in the presence of keratinocytes. Tspan8+ cells without co-culture with keratinocytes also express MMP-9 but they are unable to activate it, indicating that keratinocytes are required not only to provide proMMP-9.  The author should explain this in more detail.

The Reviewer # 3 is right to point out that proMMP-9, was also detected when melanoma cells were cultured on dermis in the absence of keratinocytes, without noticeable active MMP-9 and independently of Tspan8 expression. Nevertheless, it is important to notice that the proMMP-9 level is very low compared to those of skin reconstructs (see Figure 2 c, d). This is in agreement with the fact that keratinocytes are the main producers of proMMP-9 (Figure 6b). We have taken into account this comment and clarified it by modifying the sentence accordingly (lanes 130-132 of the revised version).

For information, we previously noticed that the ability of melanoma cells expressing or not Tspan8 to produce proMMP-9 is closely dependent from the interaction with matrix components. Indeed, when metastatic (Tspan8+) or non metastatic cells (Tspan8-) were cultured as monolayers on plastic, they produced exclusively proMMP-2, whereas when cultured on collagen IV-coated plastic, they become able to produce proMMP-9, at very low levels close to those obtained when they are cultured alone on dermis (see Figure below). However, given the primary focus of our study, we evaluate that these results are not essential for the message of our manuscript, especially since MMP-9 is known to be not constitutively produced by tumor cells. However, they could be included if the Reviewer 3 thinks it is necessary.

2.4 Tspan8 expression in melanoma cells surrounded with keratinocytes promotes proMMP-9 activation by increasing the amount of active MMP-3 and decreasing TIMP-1 levels.

Here the authors correlated the fully activation of MMP3 which is produced by keratinocytes, with co-culture with Tspan8+ melanoma cells. However, the claim that Tspan8 decreases the levels of TIMP-1 is not supported by the results. Since the co-culture is in a 1:80 ratio (melanoma cell:keratinocyte), how the authors differentiate between downregulation of TIMP-1 and dilution with low expressing keratinocyte? Additional data need to be provided to support their claim.

Overall the manuscript was well written and I think it will be of interest for the scientific community.

We are sorry to have forgotten to specify in the Materials and Methods that in the 4 types of culture that we developed, we systematically used the same number of keratinocytes (450.000 cells) and melanoma cells (5620 cells) seeded into a stainless-steel ring deposited on the surface of human dead de-epidermized dermis (DED) squares. Moreover, whatever the type of culture used, the dermis are deposited in the same petri dishes and with the same total volume of culture medium. These points have been specified in the Materials and Methods (section Invasion assay in human skin reconstructs (lines 400-404).

Because TIMP-1 is mainly produced by melanoma cells, we could compare TIMP-1 levels between cultures III and culture IV. It is why we conclude that: “ Overall, our data show that Tspan8+ melanoma cells surrounded by keratinocytes maintained very low TIMP-1 level ”. Nevertheless, it is true that we could not compare cultures III and IV with culture II. Therefore, we rewrote this sentence as follows: « Overall, our data show that Tspan8+ melanoma cells surrounded by keratinocytes maintained lower TIMP-1 level when compared to melanoma cells juxtaposed without contacts with keratinocytes. » (lines 238-239 of the revised manuscript)

Round 2

Reviewer 2 Report

The authors should provide high quality images in the final manuscript version (e.g. Figure 2B needs improvement)